# Maternal Linoleic Acid Overconsumption Alters Offspring Gut and Adipose Tissue Homeostasis in Young but Not Older Adult Rats

**DOI:** 10.3390/nu12113451

**Published:** 2020-11-11

**Authors:** Justine Marchix, Charlène Alain, Sandrine David-Le Gall, Luis Alberto Acuña-Amador, Céline Druart, Nathalie M. Delzenne, Frédérique Barloy-Hubler, Philippe Legrand, Gaëlle Boudry

**Affiliations:** 1Laboratoire de Biochimie et Nutrition Humaine, Agrocampus Ouest, 35000 Rennes, France; justinemarchix@gmail.com (J.M.); philippe.legrand@agrocampus-ouest.fr (P.L.); 2Institut NuMeCan, INRAE, INSERM, University Rennes, 35000 Rennes, France; charlene.al@hotmail.fr (C.A.); sandrine.legall-david@univ-rennes1.fr (S.D.-L.G.); 3Institut de Génétique et Développement de Rennes, CNRS, UMR6290, Université de Rennes 1, 35000 Rennes, France; luisalberto.acuna@ucr.ac.cr (L.A.A.-A.); frederique.hubler@univ-rennes1.fr (F.B.-H.); 4Metabolism and Nutrition Research Group, Louvain Drug Research Institute, UCLouvain, Université catholique de Louvain, B-1200 Brussels, Belgium; celine.druart@a-mansia.com (C.D.); nathalie.delzenne@uclouvain.be (N.M.D.)

**Keywords:** *n*-6 PUFA, gut microbiota, gut permeability, adipose tissue, conjugated linoleic acids

## Abstract

Maternal *n*-6 polyunsaturated fatty acids (PUFA) consumption during gestation and lactation can predispose offspring to the development of metabolic diseases such as obesity later in life. However, the mechanisms underlying the potential programming effect of *n*-6 PUFA upon offspring physiology are not yet all established. Herein, we investigated the effects of maternal and weaning linoleic acid (LA)-rich diet interactions on gut intestinal and adipose tissue physiology in young (3-month-old) and older (6-month-old) adult offspring. Pregnant rats were fed a control diet (2% LA) or an LA-rich diet (12% LA) during gestation and lactation. At weaning, offspring were either maintained on the maternal diet or fed the other diet for 3 or 6 months. At 3 months of age, the maternal LA-diet favored low-grade inflammation and greater adiposity, while at 6 months of age, offspring intestinal barrier function, adipose tissue physiology and hepatic conjugated linoleic acids were strongly influenced by the weaning diet. The maternal LA-diet impacted offspring cecal microbiota diversity and composition at 3 months of age, but had only few remnant effects upon cecal microbiota composition at 6 months of age. Our study suggests that perinatal exposure to high LA levels induces a differential metabolic response to weaning diet exposure in adult life. This programming effect of a maternal LA-diet may be related to the alteration of offspring gut microbiota.

## 1. Introduction

In the past three decades, the overconsumption of vegetable oils rich in *n*-6 polyunsaturated fatty acids (PUFA) and the lower intake of *n*-3 PUFA has resulted in an imbalance in the *n*-6/*n*-3 ratio, rising from 1:1 to 20:1 nowadays [1]. Consequently, the levels of linoleic acid (LA), an essential *n*-6 fatty acid, of women’s milk also increased from 6% to a plateau at around 16% of total fatty acids [2]. Strong evidence indicates that maternal nutrition during gestation and lactation can profoundly affect offspring health and disease risk later in life [3]. Specifically, LA, and especially its derivative, arachidonic acid (ARA), display pro-adipogenic properties that are related to a greater risk of obesity and related metabolic conditions [4,5]. A high intake of *n*-6 PUFA during this critical window of development results in a progressive accumulation of body fat across generations [2,6,7]. Similarly, Rudolph et al. reported that a low perinatal *n*-6/*n*-3 PUFA ratio exposure predisposed adult offspring to a phenotype of resistance to diet-induced obesity [8]. In epidemiological studies, the level of LA, of its derivatives, or of the *n*-6/*n*-3 ratio in maternal or cord plasma was positively correlated to adiposity immediately at birth (Growing up towards Healthy Outcomes (GUSTO) cohort [9]), or in childhood at 6 or 7 years of age (Generation R cohort [10], Maastrich Essential Fatty Acid Birth (MEFAB) cohort [11], Southampton Women’s Survey (SWS) cohort [12], US pregnant cohort [13]). However, this effect does not seem to be long-lasting, since no such correlation was found when offspring were 23 years old in the MEFAB cohort [14], or at 5 years in the GUSO cohort [9]. Breast milk PUFA levels have also been correlated to adiposity in two studies reporting higher fat mass at 4 months of age in the quartile with the greater ARA/*n*-3 long chain (LC)-PUFA ratio in breast milk [8], and a negative correlation has been reported between fat mass at 7 years of age and breast milk docosahexaenoic acid (DHA) content [15]. Interestingly, in several studies, maternal dietary *n*-3 PUFA supplementation did not affect offspring adiposity [16,17], suggesting that lowering *n*-6 PUFA content may be more effective in reducing the risk of metabolic disease in offspring. 

The mechanisms underlying the potential programming effect of *n*-6 PUFA status on adiposity are not all yet established. Epigenetic mechanisms have been shown to play a major role in neonatal programming by imprinting gene expression [18]. However, the microbiota may also be involved. It is well established now that host physiology, metabolism and behavior may be affected by the gut microbiota [19,20]. More specifically, altered gut homeostasis has been linked to adipose tissue expansion and inflammation through several mechanisms—increased intestinal permeability-promoting endotoxemia, modulation of energy harvested from diet by the microbiota, and altered production of bacterial metabolites (e.g., short chain fatty acids) interacting with adipocyte metabolism through G-protein coupled receptor (GPR)41/43 receptors [21,22,23,24]. This has been elegantly demonstrated by Cox et al. using antibiotic treatment—mice receiving antibiotics during the suckling period exhibited increased fat mass gain under a high-fat diet (HFD) later in life compared to vehicle-treated mice [25]. Thus, maternal nutrition and diet-modulated maternal microbiota [26] can affect offspring microbiota composition. Recent data have drawn attention to the effect of PUFA intake on gut homeostasis [27]. Specifically, an enrichment of HFD with *n*-6 PUFA has been shown to cause increased intestinal inflammation and barrier dysfunction. These alterations were associated with intestinal dysbiosis led by a blooming of the Enterobacteriaceae and Firmicutes families [28,29,30,31]. Similarly, the capacity of Fat-2 mice to endogenously overproduce *n*-6 PUFA led to a higher risk of metabolic syndrome, inflammation, bacterial translocation and non-alcoholic fatty liver, likely mediated by altered host –microbiota interactions [31]. On the other hand, studies using transgenic Fat-1 mice demonstrated that inflammation and alteration of the gut microbiota induced by a high-*n*-6, high-fat diet could be restored by the transgenic conversion of tissue *n*-6 to *n*-3 PUFA levels, suggesting a strong involvement of *n*-6 PUFA levels in gut homeostasis [30,31,32]. These studies using Fat-1 and Fat-2 mice elucidated the role of *n*-6 PUFA status, but not of dietary *n*-6 PUFA consumption, which might have a different effect on gut barrier function and microbiota. Thus, and contrary to *n*-3 PUFA, where few studies evaluated their impact on gut microbiota when incorporated into a normal/moderate fat diet [33,34,35,36], the precise role of dietary *n*-6 PUFA in intestinal homeostasis is difficult to identify due to the confounding factor of high calorie intake, differences in type and source of PUFA, duration of intervention or age of individuals. This is even more true when examining the effect of maternal dietary *n*-6 PUFA in a moderate fat diet during gestation and lactation on host–microbiota interactions. In a recent study, Shrestha et al. demonstrated that a high consumption of LA (6.2% of energy) in a normal fat diet (9% of energy) does not significantly alter the gut microbiota composition of dams during pregnancy as compared to a lower intake of LA, but they did not investigate the offspring microbiota [37].

Thus, taking into account the possible impact of *n*-6 PUFA on adiposity, gut barrier function and the importance of the microbiota in neonatal programming, our study aimed at evaluating the impact of maternal *n*-6 PUFA intake in a moderate fat (21% energy) diet during gestation and lactation on offspring gut homeostasis and adipose tissue physiology, in young (3-month-old) and older (6-month-old) adult rats. 

## 2. Materials and Methods 

### 2.1. Animal Protocol

All experiments were performed in accordance with the European Union Guidelines for animal care and use (2003/35/CEE) and approved by the Ethics & Animal Experimentation Committee of Rennes (MENESR under number: 01375.02).

The control diet (C-diet) and the LA-rich diet (LA-diet) were isocaloric and isolipidic. Both diets contained 10.0% fat (21% of total energy), 22.0% proteins, 70.3% carbohydrates, 2.0% fiber and 5.7% minerals and vitamins. We chose a moderate fat diet (21% of energy) in order to allow for high proportions of LA intake without reaching the fat percentage observed in high-fat diets. The lipid blends were made from a combination of commercial vegetable oils, C16:0 triglycerides and ethyl LA (TCI Europe, Zwijndrecht, Belgium) at the Unité de Production d’Aliments Expérimentaux (INRA, Jouy en Josas, France). The fatty acid composition of diets was confirmed by gas chromatography (Appendix A). Experimental diets were stored at –20 °C and provided fresh every day. 

Eight-week-old female Wistar rats from Janvier Labs (Le Genest-Saint-Isle, France) were fed either the C- or LA-diets during pregnancy and lactation (*n* = 4–5 per group). At birth, pups were culled to 8 per litter. At weaning, on postnatal day 21, male offspring were randomly separated into two groups (10 rats per group). Pups were maintained on the same maternal diet or fed the other experimental diet, thereby generating 4 groups that differed from maternal and/or weaning diets (C-C, C-LA, LA-C and LA-LA, Appendix A). All animals were housed on a 12 h light/dark cycle and maintained at 22 ± 2 °C with free access to water and food ad libitum. They were fed their respective weaning diets for 3 (*n* = 4 rats/dietary group, except for C-LA, *n =* 3) or 6 (*n =* 6/dietary group) months. Within a dietary group, rats were housed in 5 different cages with 2 rats/cage.

Body weight was measured every other day. At weaning, the dams and some of the remaining pups (*n =* 3 dams/dietary group and *n =* 3 pups/dietary dam group) were euthanized to collect liver and epididymal tissue as well as mammary glands from the dams. Tissues were immediately snap-frozen for later fatty acid composition determination. At the end of the 3- or 6-month period, fasted rats were anesthetized with an intraperitoneal injection of pentobarbital (140 mg/kg, Merial, Lyon, France) and blood samples were collected by cardiac puncture into EDTA K2-treated vacutainers (Dominique Dutscher, Brumath, France). Plasma was obtained by centrifugation (3000× *g*, 15 min, 4 °C) and stored at −20 °C until further analysis. Adipose tissues were dissected out and weighted. Epididymal adipose tissue samples were washed and fixed in 4% paraformaldehyde (1 week) for further histology analysis or snap-frozen for fatty acid composition determination. Other samples of epididymal adipose tissue were snap-frozen in liquid nitrogen and stored at −80 °C until RNA extraction. Cecal contents were collected, snap-frozen and stored at −80 °C. Cecal and colonic tissue were rinsed with ice-cold phosphate buffer saline (PBS). Colonic tissue was snap-frozen and stored at −80 °C until further analysis, while cecal tissue was stored in ice-cold Dulbecco modified Eagle’s minimal essential medium (DMEM) for immediate mounting in Ussing chambers. 

### 2.2. Microbiota Analysis

Bacterial DNA was extracted from 100–150 mg of cecal luminal content samples using the ZR fecal DNA Miniprep kit (Zymo Research, USA). The V3-V4 region of 16S rRNA gene was amplified using the following primers: CTTTCCCTACACGACGCTCTTCCGATCTACTCCTACGGGAGGCAGCAG (V3F) and GGAGTTCAGACGTGTGCTCTTCCGATCT TACCAGGGTATCTAATCC (V4R), Taq Phusion (New England Biolabs) and dNTP (New England Biolabs) during 25 cycles (10 s at 98 °C, 30 s at 45 °C, 45 s at 72 °C). The purity of amplicons was checked on agarose gels before sequencing using Illumina Miseq technology, performed at the Genotoul Get-Plage facility (Toulouse, France). Briefly, single multiplexing was performed using a homemade 6 bp index, which was added to R784 during a second PCR in 12 cycles using forward primer (AATGATACGGCGACCACCGAGATCTACACTCTTTCCCTACACGAC) and reverse primer (CAAGCAGAAGACGGCATACGAGAT-index-GTGACTGGAGTTCAGACGTGT). The resulting PCR products were purified and loaded onto the Illumina MiSeq cartridge according to the manufacturer’s instructions. The quality of the run was checked internally using PhiX, and then each paired-end sequence was assigned to its sample with the help of the previously integrated index. Each paired-end sequence was assembled using Flash [38], using at least a 10 bp overlap between the forward and reverse sequences, allowing 10% mismatch. The lack of contamination was checked with a negative control during the PCR (water as template). The quality of the stitching procedure was controlled using 4 bacterial samples that ran routinely in the sequencing facility in parallel with the current samples. Raw sequences can be found at https://data.inrae.fr/dataset.xhtml?persistentId=doi:10.15454/Y2AHF7. Raw sequences were analyzed using the bioinformatic pipeline FROGS [39], based on several R software [40] packages. Briefly, quality control and trimming was performed using Cutadapt (version 1.18) [41] and Flash (version 1.2.11) [38] to keep sequences between 380 and 500 bp, sequences with ambiguous bases and sequences that do not contain good primers. Clustering was then performed using Swarm v2.1.2 [42] with an aggregation maximal distance of 3 bases. Chimera were removed using VSEARCH (v2.9.1) [43] with cross-sample validation. PhiX contamination was removed using VSEARCH. Affiliation was performed using the Silva 123 16S database [44] and NCBI Blastn++ [45]. A phylogenetic tree was constructed using FastTree (v2.1.10) [46] and sample depth was normalized using GMPR [47].

The phylosseq package [48] was used for biostatistical process. Alpha-diversity was estimated using two metrics: the number of observed species (richness index) and the Shannon index. A two-way ANOVA test was then performed on these two indexes to assess maternal and weaning diet effects and their interaction. Cage effect was not considered in the statistical analysis. Beta-diversity was evaluated by calculating the weighted Unifrac distance between samples. Ordination using principal coordinates analyses (PCoA) was performed to represent samples on a 2D plot. Multivariate ANOVA was performed using adonis function (Vegan Package, https://cran.r-project.org, https://github.com/vegandevs/vegan) with 9999 permutations. Relative abundances of the major phyla and of families were compared using two-way ANOVA to test maternal, dietary and their interaction effects.

### 2.3. Intestinal Alkaline Phosphatase Activity (IAP) and Ussing Chamber Assay

IAP activity in the colon was determined using the Sensolyte p-nitrophenyl phosphate Alkaline Phosphatase Assay kit (Anaspec, Fremont, CA, USA) and expressed as specific activity (Arbitrary Unit/mg of protein) after protein quantification in tissue homogenates (Pierce Protein BCA Protein Assay kit, ThermoFisher Scientific). Cecal tissues were mounted into Ussing chambers to measure permeability to probes of various size (FITC-dextran 4000 (FD4, 4 kDa) and horseradish peroxidase (HRP, 40 kDa) as well as tissue conductance as previously described [49]. 

### 2.4. Real-Time PCR 

Total RNA from colonic and epididymal adipose tissue was extracted via the Trizol method (15596-018; Fischer Scientific) and quantified using a spectrophotometer (Denovix). In total, 2 μg RNA was converted to cDNA using a High Capacity Complementary DNA Reverse Transcription Kit (Applied Biosystems) following the manufacturer’s protocol. Real-time PCR was performed with a StepOnePlus real-time PCR machine using SyberGreen master mix (Fischer Scientific) for detection. Primers for selected genes (Appendix A) were designed using Integrated DNA Technologies Primer Quest. HPRT-1, RPS18 and Actin were used as housekeeping genes. The standard curve method was assessed for each gene and the 2^−ΔΔCt^ method was applied for quantification. The reference and targeted gene primers had efficiencies between 95 and 101%, (R^2^ ≥ 0.98). Data were normalized to the C-C group for each respective age as a reference group. 

### 2.5. Adipose Tissue Histology

Epididymal fat pads were fixed in 4% paraformaldehyde (PAF), processed and embedded in paraffin using a tissue processor (Excelsior ES; ThermoFischer Scientific, Waltham, MA, USA) for histology. Sections of 5 µm thickness were deparaffined and stained with hematoxylin and eosin for morphometric analysis, and then scanned with the NanoZoomer 2.0 RS (Hamamatsu, Tokyo, Japan). Images were captured at 20× magnification for analysis. Five representative pictures of each section were selected for counting. A semi-automated custom image analysis protocol was developed using ImageJ software to quantify the area of individual adipocytes. Histological images were segmented, creating a binary mask and converting the image to 1-bit configuration. The adipocytes were counted if they met the following criteria: (1) the adipocyte has an equivalent sphere surface diameter between 20 and 200 µm; (2) the adipocyte has a shape factor of 0.5-1 (circularity indicator) and (3) the adipocyte does not border the image frame. Over 10% of the adipocytes were included manually. Each adipocyte was subsequently labeled with a number to visualize the adipocytes more clearly and provide traceability for their respective quantified area. About 100 adipocytes were counted per picture. Morphometric data were then exported, and expressed as adipocyte frequency per mean diameter for statistical analysis.

### 2.6. Fatty Acid Analysis and Conjugated Linoleic Acid Quantification

Solvents used in the lipid analysis were purchased from Fischer Scientific (Elancourt, France). Lipids were extracted as previously described [50]. Briefly, lipids were extracted from adipose tissue, saponified and methylated. Fatty acid methyl esters (FAME) were extracted and analyzed by gas chromatography with a flame ionization detector (Agilent Technologies 6890N, Bios Analytique, France). Hydrogen was used as the carrier gas. The identification of FAME peaks was based on the retention times obtained from fatty acids standards. The area under peaks was determined using ChemStation software (Agilent), and results were expressed as the total fatty acid percentage. Conjugated linoleic acids (CLA) were quantified in the liver as already described [51].

### 2.7. Statistical Analysis

Data are presented as box and whisker plots with means and min-max. At each age (3 and 6 months), the data (body weight, adiposity, gut function, qPCR data) were analyzed by two-way ANOVA testing of the maternal diet, weaning diet and maternal × weaning diet effects using the GraphPad Prism software. For adipocyte distribution analysis, Gaussian curves were fit to each dataset and a comparison of fits was performed. At weaning (dams’ and pupss tissue fatty acid composition), data were analyzed by Mann–Whitney tests. Microbiota data were analyzed as described in the corresponding session. Spearman rank correlation between data was performed using cor function in R and the corrplot R package of R. 

## 3. Results

### 3.1. Maternal LA-Diet Impacted Dam and Offspring Tissue Composition at Weaning

We verified that the maternal diet affected maternal tissue fatty acid composition as well as offspring tissue fatty acid composition at weaning. At the end of the gestation and lactation periods, although dam body weight was similar (C: 329 ± 7 vs. LA: 313 ± 10 g, *p* > 0.05), the LA-diet impacted the fatty acid compositions of multiple tissues. Indeed, fatty acid analyses of dam liver, epididymal adipose tissue and the mammary gland indicated the maternal diet intake of greater proportions of *n*-6 PUFA (LA and ARA) in all tissues at the expense of mono-unsaturated fatty acids (MUFA) in the LA-fed animals (Appendix A). At weaning, offspring tissue fatty acid composition was also profoundly impacted by the maternal diet with greater proportions of *n*-6 PUFA and a lower proportion of MUFA in the liver and epididymal adipose tissue (Appendix A) despite similar body weights (C: 60 ± 5 vs. LA: 63 ± 4 g, *p* > 0.05). 

### 3.2. Maternal LA-Diet Impacted Gut Barrier Function and Adipose Tissue in Young Adult Offspring 

#### 3.2.1. Cecal Barrier Function and Inflammation 

At 3 months of age, cecal barrier function was influenced by the maternal and/or the weaning diets. Compared to C-C rats, the cecal permeability to small molecules (FD-4) was greater in rats exposed to the LA-diet either through the maternal or the weaning diet, i.e., C-LA, LA-C and LA-LA groups, (maternal diet × weaning diet, *p* = 0.03, Figure 1A). The cecal permeability to large molecules (HRP) was greater in C-LA rats compared to those fed the weaning C-diet (maternal diet × weaning diet, *p* = 0.04, Figure 1B). Cecal conductance was greater in rats fed the weaning LA-diet (weaning diet *p* = 0.01, Figure 1C). We also measured IAP activity. IAP is a brush border enzyme secreted along the intestine with several biological roles, including LPS detoxification, the resolution of inflammation and the maintenance of gut microbiota homeostasis. Several reports have suggested that *n*-6 PUFA can impact intestinal production and secretion of IAP [28,30,52,53]. IAP activity in the colon was lower in rats born to LA mothers compared to those born to C-mothers, irrespective of their weaning diet (maternal diet *p* = 0.02, Figure 1D). Finally, we evaluated local and systemic inflammation. Colonic *tnf-α* mRNA levels tended to be greater in rats born to LA dams compared to those born to C ones (maternal diet *p* = 0.08, Figure 1E); *il-1β* and *mcp-1* mRNA levels were not affected by the maternal or the weaning diet (data not shown). The plasma IL-1β concentration was greater in rats born to LA mothers compared to those born to C ones, irrespective of their weaning diet (maternal diet *p* = 0.004, Figure 1F).

#### 3.2.2. Epididymal Adipose Tissue

Based on the concept of a “leaky gut” in the pathogenesis of several metabolic diseases [54], we further analyzed the impact of LA-diet on adipose tissue physiology in young offspring. We first confirmed the dietary fatty acid’s incorporation into the offspring tissue by analyzing the epididymal adipose tissue fatty acid composition. The fatty acid composition of rat epididymal fat at 3 months of age was affected by the weaning, but not the maternal diet, with greater proportions of *n*-6 PUFA at the expense of MUFA in rats fed the weaning LA-diet compared to the weaning C-diet, irrespective of the maternal diet (Table 1).

We next evaluated adipose tissue morphology and lipid metabolism. Rats born to LA mothers tended to be heavier (maternal diet *p* = 0.08, Figure 2A) than those born to C mothers, irrespective of their weaning diet. Adipose tissue physiology was altered mainly by the maternal diet, but not the weaning diet, with enhanced visceral adiposity in rats born to LA mothers (maternal diet *p* = 0.04, Figure 2B) but no difference in the percentage of subcutaneous adipose tissue (Figure 2C). When comparing the means of Gaussian curve fits of the adipocyte distribution of epididymal fat, the offspring of LA mothers exhibited an upwards shift due to a higher frequency of larger adipocyte (*p* = 0.02, Figure 2D). Adipose tissue metabolism was also affected by the maternal diet with lower *lipe* (maternal diet *p* = 0.02, Figure 2E) and *ppar-γ* (maternal diet *p* = 0.04, Figure 2F) mRNA levels in rats born to LA mothers compared to those born to C mothers, irrespective of their weaning diet. Transcriptomic expressions of *fasn* and *lpl* were not altered by the diet (Figure 2G,H). 

#### 3.2.3. Conjugated-Linoleic Acids

There are several potential mediators of the crosstalk between the gut and adipose tissues, including the CLAs. CLAs are produced from LA by bacteria in the intestinal lumen, but can transfer to the inner milieu and impact host physiology [51]. CLA concentrations were measured in the liver as a reflection of CLA production and absorption through the intestine. At 3 months of age, hepatic CLA concentrations were mainly affected by the weaning diet. Rats fed the weaning LA-diet had increased hepatic levels of 18:2 *trans*-10, *cis*-12, but reduced levels of 18:2 *cis*-9, *cis*-11, (*p* = 0.04 and 0.01, respectively, Table 2). Rats born to LA dams displayed a tendency for greater levels of 18:2 *cis*-9, *trans*-11 and 18:2 *trans*-10, *cis*-12 (*p* = 0.08 and 0.08, respectively, Table 2) in their liver compared to rats born to C dams. Higher hepatic levels of 18:1 *trans*-11 were observed in rats fed the weaning LA-diet when they were born to LA mothers (maternal diet x weaning diet *p* = 0.02, Table 2). 

### 3.3. Maternal LA-Diet Had a Limited Impact on Gut Barrier Function and Adipose Tissue in Older Adults Compared to The Weaning Diet Itself

#### 3.3.1. Cecal Barrier Function and Inflammation

At 6 months of age, no maternal diet effect was observed on cecal barrier function or inflammation. However, the weaning diet affected cecal homeostasis—FD-4 (Figure 3A) and HRP (Figure 3B) fluxes across the cecal mucosa were decreased in weaning LA-diet-fed rats, irrespective of the maternal diet (weaning diet *p* = 0.01 and 0.04, respectively). Cecal conductance was decreased by the weaning LA-diet in rats born to C mothers only (weaning x maternal diet *p* = 0.03, Figure 3C). The activity of IAP tended to be greater (weaning diet *p* = 0.07, Figure 3D), while the colonic *tnf-α* mRNA level was reduced (weaning diet *p* = 0.03, respectively Figure 3E) in rats fed the weaning LA-diet, irrespective of their maternal diet. Plasma IL-1β concentration was lower in rats fed the weaning LA-diet, irrespective of the maternal diet (data not shown, published in Marchix et al., 2020 [50]).

#### 3.3.2. Epididymal Adipose Tissue

As previously observed at 3 months of age, the fatty acid composition of rat epididymal fat at 6 months of age was mainly affected by the weaning diet, with greater proportions of *n*-6 PUFA at the expense of MUFA in rats fed the weaning LA-diet compared to the weaning C-diet, irrespective of the maternal diet (Table 3). SFA proportions was greater in rats born to LA mothers, compared to those born to C mothers (*p* = 0.05, Table 3).

Compared to 3-month-old rats, the weaning LA-diet rather than the maternal LA-diet affected adipose tissue physiology in older offspring. The weaning LA-diet did not significantly impact rat body weight (Figure 4A), however it decreased the visceral adiposity (weaning diet *p* = 0.003, Figure 4B) but not the subcutaneous one (Figure 4C). A slight maternal diet effect was still observed for visceral adiposity, which was greater in rats born to LA mothers (maternal diet *p* = 0.04, Figure 4B). Furthermore, the comparison of the means of the Gaussian curve fits of the adipocyte distribution showed the long-lasting effect of maternal LA-diet, with a higher frequency of greater adipocyte in LA-LA offspring compared to the C-LA ones (*p* < 0.01, Figure 4D). The lipid metabolism key enzyme expression in the epididymal adipose tissue was not affected by the diet (Figure 4E–H). 

#### 3.3.3. Conjugated-Linoleic Acids

At 6 months of age, the hepatic CLA concentrations were affected by the weaning diet with greater levels of 18:2 *cis*-9, *trans*-11 and 18:1 *trans*-11 (weaning diet *p* < 0.0001 and *p* = 0.02, respectively, Table 4) and lower levels of 18:2 *cis*-9, *cis*-11 and 18:2 *trans*-11, *trans*-13 (weaning diet *p* < 0.01 and *p* = 0.04, respectively, Table 4) in the liver of rats fed the weaning LA-diet, compared to the weaning C-diet-fed ones, irrespective of the maternal diet. Maternal diet affected the hepatic 18:2 *trans*-9, *trans*-11 levels, which were markedly lower in the livers of rats born to LA dams, irrespective of the weaning diet (maternal diet *p* < 0.01, Table 4). The hepatic level of 18:2 *trans*-10, *cis*-12 was not influenced by any of the diets at 6 months of age (Table 4).

### 3.4. Cecal Microbiota

Gut dysbiosis has often been associated with the development of metabolic diseases and intestinal barrier dysfunction [30]. Moreover, while it is well established that diet can alter the gut microbiota, evidence also suggests that maternal diet-induced changes in microbiota could be transferred to offspring [55,56]. Therefore, we determined whether altered cecal gut microbiota was related to changes in gut physiology in offspring at both ages. These microbiota data need, however, to be taken with caution, due to the low n number, which is not optimal for microbiota analysis [57,58].

#### 3.4.1. Cecal Microbiota at 3 Months of Age

At 3 months of age, principal coordinates analysis (PCoA) using the weighted Unifrac distance revealed the distinct clustering of cecal microbiota in the four dietary groups (PERMANOVA *p* = 0.008, Figure 5A), with a significant effect of the maternal diet (*p* = 0.04) but not of the weaning diet (*p* = 0.30). Indeed, the cecal microbiota of rats born to C and LA mothers were segregated, but only the microbiota of rats born to C mothers were clustered according to the weaning diet (Figure 5A). The cecal microbiota of rats born to LA mothers exhibited a greater richness (number of observed species) than those born to C mothers (*p* = 0.001, Figure 5B). However, the weaning LA-diet had the opposite effect, with reduced cecal microbiota richness (*p* = 0.009, Figure 5B). Microbial diversity estimated by the Shannon index also tended to be greater in rats born to LA mothers compared to those born to C mothers (*p* = 0.07, Figure 5C). At the phylum level, rats born to LA mothers had greater relative abundances of Proteobacteria (maternal diet effect, *p* = 0.03) and lower relative abundances of Firmicutes (maternal diet effect, *p* = 0.04) compared to rats born to C mothers (Figure 5D). No significant effect of the weaning diet was observed at the phylum level. To further characterize the impact of the maternal diet on the cecal microbiota’s response to the weaning diet, we analyzed the difference in family abundances between dietary groups (Appendix A). Most of the differences were attributed to the maternal diet or to an interaction between the maternal and the weaning diets. It is noteworthy that rats born to LA dams exhibited greater relative abundances of Tannerellaceae, Ruminococcaceae, Rhodospirillaceae and Desulfovibrionaceae, and lower relative abundances of Sporichthyaceae and Akkermansiaceae, irrespective of the weaning diet (Appendix A). Moreover, all the rats that had received LA (C-LA, LA-C and LA-LA groups) had reduced levels of Lactobacillaceae and Clostridiaceae compared to the C-C group (Appendix A). Moreover, C-LA rats had increased levels of Bacteroidaceae, Christensenellaceae, Lachnospiraceae and Mollicutes (Appendix A). 

#### 3.4.2. Cecal Microbiota at 6 Months of Age

While at 3 months of age the cecal microbiota were profoundly impacted by the maternal and/or weaning diets, the clustering of rat cecal microbiota upon maternal diet and/or weaning diet was less apparent on the weighted Unifrac distance PCoA at 6 months of age (Figure 6A). No significant differences were noted when comparing the microbiota compositions associated with maternal or weaning diet (PERMANOVA *p* = 0.07 and 0.21, respectively). As opposed to the 3-month-old rats, the cecal microbiota richness (number of observed species) tended to be lower (*p* = 0.07, Figure 6B) and the diversity (Shannon index) was significantly lower (*p* = 0.04, Figure 6C) in 6-month-old rats born to LA mothers compared to those born to C mothers. At the phylum level, no maternal diet effect was observed, but the relative abundances of Bacteroidetes (maternal x weaning diet *p* = 0.002) and of Firmicutes (maternal x weaning diet *p* = 0.02) were respectively lower and greater in C-LA rats compared to C-C ones (Figure 6D). At the family level (Appendix A), the maternal LA-diet had some slight yet significant effects on the relative abundances of several families (decreased abundance of Muribaculaceae, increased abundances of Micrococcaceae, Lactobacillaceae and Erysipelotrichaceae (Appendix A)). The weaning LA-diet slightly reduced the abundance of Lachnospiraceae, and some maternal x diet interactions were still observed, such as for Tannerellaceae abundance (Appendix A). 

Interestingly, when we compared the cecal microbiota between 3- and 6-month-old rats, the cecal microbiota of rats born to LA mothers exhibited only a few changes between 3 and 6 months of age compared to rats born to C mothers, which exhibited marked changes with age. This was demonstrated by weighted Unifrac distance PCoA (Appendix A) and the absence of increase in richness and diversity with age in rats born to LA mothers, irrespective of the weaning diet, compared to rats born to C mothers (Appendix A). Likewise, the relative abundances at the family level did not changed markedly with age in rats born to LA mothers compared to those born to C mothers (Appendix A). 

#### 3.4.3. Microbiota Composition Correlates with Rat Gut Barrier and Obesity Phenotype at 3 but Not 6 Months of Age

Spearman rank correlations between microbiota family relative abundances, hepatic CLA concentrations, key gut barriers (FD4 and HRP fluxes, G, IAP activity) and obesity and low-grade inflammation (body weight, visceral adiposity index, plasma IL-1β) parameters were performed to highlight possible links between changes in microbiota and rat phenotype. 

At 3 months of age, a cluster of bacterial families, including Peptococcaceae, Enterococcaceae, Lactobacillaceae, Clostridiaceae, Bifidobacteriaceae, and to a lesser extent the Rikinellaceae and Clostridiaceae vadinBB60 groups, correlated negatively with both gut barrier parameters and obesity/inflammation markers (Figure 7A). Two other bacterial clusters correlated with gut barrier parameters: the relative abundances of Tannerellaceae and Rhodospirillaceae correlated negatively with IAP activity, and the relative abundances of Bacteriodeceae, Desulfovibrionaceae, Mollicutes RF39, Eubacteriaceae, Burkholderiaceae and Enterobacteriaceae correlated positively with gut permeability parameters. Hepatic 18:2 *cis*-9, *cis*-11 correlated negatively with gut permeability parameters, and 18:1 *trans*-11 with body weight and adiposity (Figure 7A).

At 6 months of age, less significant correlations were observed (Figure 7B). However, the relative abundances of Bacteroideaceae and Peptococcaceae correlated negatively with gut permeability parameters. Enterobacteriaceae relative abundances correlated positively with IAP activity. Visceral adiposity correlated with the hepatic levels of many CLAs—positively with 18:2 *trans*-11, *trans*-13 levels and negatively with 18:2 *cis*-9, *trans*-11, 18:2 *trans*-9, *trans*-11 and 18:1 *trans*-11 levels. Cecal permeability correlated positively with hepatic 18:2 *trans*-11, *trans*-13 levels and negatively with 18:2 *cis*-9, *trans*-11 and 18:1 *trans*-11 levels (Figure 7B).

## 4. Discussion

In this study, we investigated the interactive effects of high LA intake in the maternal diet during gestation and lactation and in the weaning diet on intestinal and adipose tissue and gut microbiota, in young (3-month-old) and older (6-month-old) adult offspring. To our knowledge, our report is the first to demonstrate the effects of such diet interaction on host physiology and its associated alterations of gut microbiota, in the context of moderate fat intake. Here, we found that exposure to LA during gestation and lactation had long-lasting effects on offspring gut functions at 3 months of age. Maternal high LA intake appeared to be responsible for the major changes in inflammation with increased colonic *tnf-α* mRNA and plasma IL-1β, and a decreased IAP activity. IAP has been well described for its role in LPS detoxification and in tissue resolution of inflammation, but a role in the maintenance of gut microbiota homeostasis has also been suggested [59,60]. The reduced IAP activity induced by maternal LA-diet in offspring gut at 3 months of age corresponds with changes in pro-inflammatory cytokines, and may be involved in gut barrier function and gut microbiota alterations. Our study supports a recent multi-omics study using transgenic mice able to overproduce *n*-6 PUFA, which suggested the potential harmful effects of *n*-6 PUFA excess, with increases in endotoxemia and inflammation. These *n*-6 PUFA effects likely resulted from an altered gut microbiota with a dominance of pro-inflammatory bacteria, such as Proteobacteria [31], as also observed in our study of rats born to LA mothers. According to our study, the authors observed a lower IAP activity in wild type mice with a high *n*-6 status compared to Fat-1 mice with high *n*-3 status, and suggested IAP to be a mediator of the effects of *n*-6/*n*-3 tissue composition on gut microbiota [30]. On the other hand, intestinal permeability was mainly altered by high LA in the weaning diet, in line with previous studies [31,61,62]. Using the intestinal cell line, Jiang et al, demonstrated that LA reduced transepithelial cell resistance (TER), a measure of barrier integrity, and increased paracellular permeability [61]. Kirpich et al. also showed that a high intake of *n*-6 PUFA exacerbates the alcohol-induced increase in permeability in a model of alcoholic liver disease [62]. Thus, our results suggest that the inflammatory process in offspring might be imprinted by the maternal diet-induced gut microbiota alterations, while intestinal barrier function might be driven mostly by the weaning diet. 

The mechanisms by which LA modulates intestinal barrier function remain to be fully determined, but several reports have suggested that CLAs could participate in these changes. CLAs naturally occur in food sources, but several in vitro and in vivo studies have demonstrated the ability of specific gut bacteria species to convert LA into diverse CLA isomers [51,63]. Here, hepatic CLA levels, which reflect gut CLA production and absorption, were mainly affected by the weaning diet. The levels of 18:2 *cis*-9, *cis*-11 were reduced in rats fed the weaning LA-diet, consistent with those of LA-rich oil force-fed mice [51]. Furthermore, 3 months of weaning LA-diet led to an increase in rumenic acid (RU, 18:2 *cis*-9, *trans*-11) and 18:2 *trans*-10, *cis*-12, the major isomers of CLAs also observed after gavage with LA rich oil [51]. The LA weaning diet led to an increase in the abundance of some genera of the Lachnospiraceae family, including the genera *Blautia* and *Roseburia*, in the offspring of C mothers. Lachnospiraceae and *Roseburia* bacterial genera have been previously described to participate in the biohydrogenation of LA into CLAs [51,63]. Thus, difference in microbiota composition could account for this difference in the major CLA isomers produced upon age and diet. We speculate that in addition to a direct effect of LA on intestinal barrier integrity, the higher levels of 18:2 *trans*-10, *cis*-12 in weaning LA-diet-fed offspring might also contribute to the alteration of the intestinal function at 3 months of age. Previous work already supports this hypothesis [64]. Specifically, the supplementation of Caco-2 cells with 18:2 *trans*-10, *cis*-12 alters the distribution of tight junction-associated proteins, occludin and ZO-1, involved in the maintenance and integrity of the epithelial barrier. This was accompanied with a greater paracellular permeability to [^14^C] mannitol and a delay in the increase of TER [64]. Negative correlations between gut permeability and specific bacterial families also suggest that the decrease in the abundance of some beneficial species, such as *Lactobacillus* [65], *Enterococcus* [66] or *Bifidobacterium* [65], may contribute to the weaning LA-diet-induced gut barrier dysfunction.

High LA intake can increase the levels of CLAs, which mainly accumulate in the cecum and colon and act locally [51,63], but some studies have also demonstrated a CLA enrichment in peripheral tissues following CLA supplementation [67]. This suggests that CLAs could exert both local and systemic effects on host physiology. At 3 months of age, we found that changes in gut barrier function and inflammation were associated with alterations of adipose tissue physiology in rat offspring. These changes were mainly driven by the maternal LA diet. The offspring of mothers fed the LA-diet during gestation and lactation exhibited higher visceral fat mass, larger adipocyte frequency and reduced hormone-sensitive lipase transcript level. This is in line with previous studies suggesting an effect of LA-enriched diet on greater adiposity in both animal and humans [6,7,10]. However, the lower expression of PPARγ transcripts in the offspring of LA mothers contrasts with previous work showing that PPARγ is a key mediator of adipogenesis induced by LA-derivative fatty acid [2,6], and may suggest another adiposity-induced mechanism in our study. Furthermore, it was later determined that CLAs could alter adipose tissue physiology (for review [68]). While the 18:2 *trans*-10, *cis*-12 isomer has been described as the main isomer responsible for reducing adiposity [69], the increased levels of 18:2 *trans*-10, *cis*-12 observed in 3-month-old offspring in the present study do not seem to corroborate the literature. In our study, CLA level variation appeared to depend mainly on the weaning diet at 3 months, while adiposity changes were related to maternal diet and, therefore, may suggest that adipose tissue changes might be related to other mechanisms or other LA derivatives. Furthermore, the negative correlations of some bacterial families with both gut barrier, plasma IL-1β and adipose tissue parameters suggests a role for gut-mediated systemic low-grade inflammation in the alteration of adipose tissue physiology [22], which warrants further investigation. 

At 6 months of age, these alterations were not exacerbated as expected, but rather showed opposite results. Our data provide evidence that the maternal LA dietary level impacts gut function, adipose tissue physiology and gut microbiota in young adult rats, but that this imprinting effect fades later in life in favor of a direct effect of the weaning diet, as previously suggested [65]. Microbiota composition seems to be more similar within the offspring groups, although microbiota richness and diversity and the abundance of some families were still altered by the maternal LA-diet. Unlike the 3-month-old offspring, rats fed the weaning LA-diet for 6 months displayed reduced intestinal permeability and inflammation (IAP, colonic *tnf-α* mRNA, plasma IL-1β, etc.), as well as lower adiposity, compared to those fed the weaning C-diet. These alterations may result from the increased levels of RU and 18:2 *trans*-9, *trans*-11 and lower levels of 18:2 cis-9, *cis*-11 and 18:2 *trans*-11, *trans*-13 CLA isomers, since visceral adiposity correlated significantly with these CLA isomer hepatic levels at 6 months. The anti-adiposity properties of CLAs have been mainly attributed to 18:2 *trans*-10, *cis*-12, but this CLA was no longer increased at 6 months of age. The beneficial effects of CLA isomers have been mainly described in relation to the CLA mixture of RU and 18:2 *trans*-10, *cis*-12, but the specific role of each isomer is still unclear [68]. In addition to CLAs, the biohydrogenation pathway leads to other intermediates, among which vaccenic acid (VA, *trans*-11 18:1) is the most important and could also be associated with beneficial effects on the host [70]. Here, we found that VA was significantly increased in offspring fed weaning LA-diets over 6 months. This is in line with previous studies showing that VA attenuates complications in metabolic syndrome, including low-grade inflammation, by acting directly in the intestine and adipose tissue [71,72]. Thus, physiological alterations of gut and adipose tissue may also result from the combined variations of the other CLAs, but the isomer-specificity of CLAs needs to be further elucidated. Interestingly, unlike the 3-month age, lower levels of the CLA isomer 18:2 *trans*-9, *trans*-11 were observed in the offspring of dams exposed to the LA-diet during gestation and lactation, but the biological meaning of this remains to be fully explored [73]. Thus, at 6 months of age, LA weaning diet-fed rats appeared to exhibit a healthier phenotype compared to C weaning diet-fed ones, with lower cecal permeability and colonic and plasma cytokine levels, as well as reduced adiposity. However, recent published data from our group revealed a hepatic steatosis in these animals [50], suggesting the strong alteration of lipid and glucose metabolism in these animals. Hepatic steatosis was associated with increased levels of the LA derivates oxidized linoleic acid metabolites (OXLAMs) and pro-inflammatory eicosanoids. Further studies are required to identify the balance among LA derivates, LA-induced changes in gut microbiota, and their specific effects on host physiology.

The gut microbiota are considered a major regulator of host physiology and metabolism. Alterations of intestinal microbiota have been described in several chronic metabolic diseases, including obesity, in which the concept of altered gut permeability and adipose tissue plasticity have been described (for review [54]). Although the number of animals studied was low and caution needs to be taken with these results [57,58], we observed a long-lasting effect of the maternal LA-diet on the gut microbiota’s composition and diversity compared to the effects of a direct weaning LA-diet in young adult offspring. Indeed, microbiota composition and diversity at 3 months of age differed between rats born to C or LA mothers. Moreover, the weaning LA-diet strongly impacted the microbiota composition when rats were born to C mothers (C-LA group compared to C-C group), but not when born to LA mothers (LA-LA group compared to LA-LA group). Interestingly, all the 3-month-old rats exposed to LA exhibited reduced Lactobacillaceae and Clostridiaceae relative abundances when compared to the C-C group, suggesting that these two families are sensitive to LA. Both families were negatively correlated to gut permeability and obesity/inflammatory parameters. At 6 months of age, some remnant effects of the maternal diet were still observed, and the weaning diet had almost no impact on the gut microbiota. Our data are in line with previous studies showing the long-term impacts of maternal HFD on offspring gut microbiota composition in animal models [74,75] or in humans [76], and the long-term impact of maternal *n*-3 PUFA intake [34,77]. However, in these studies, the offspring were usually kept on a control diet at weaning. If offspring were challenged with an experimental diet, the maternal diet effect disappeared or faded [78,79,80]. For example, in macaques, an effect of maternal HFD on fecal microbiota was observed 4 months after weaning in animals kept on a control diet, but this maternal diet effect was no longer observed when offspring were fed an HFD after weaning [80]. Interestingly, the same group published later on that maternal HFD impacted microbiota maturation throughout adult life—the offspring of HFD mothers reached a mature microbiota as early as 1 year of age, whereas the offspring of control mothers reached this stage 2 years later [81]. Our data also suggest a more ‘mature’ microbiota in rats born to LA mothers, with few changes in composition, richness and diversity between 3 and 6 months of age in these animals. The establishment and maturation of the gut microbiota are under dynamic processes, and can be influenced by various environmental factors, including the diet [82]. Our work suggests that high LA intake during gestation and lactation may account for the different colonization patterns of offspring microbiota. Interestingly, while high gut microbiota diversity is often associated with a healthy host phenotype in adults [83], this might not be the case in younger individuals. For example, breast-fed infants harbor lower fecal microbiota richness than formula-fed ones [84], yet breast-feeding is associated with lower risks of diseases. Further analysis will be required to fully determine the causal effects of maternal LA on precocious microbiota maturation and its impact on host phenotype. 

Our study has several limitations that we need to acknowledge. As already mentioned, the number of animals per group may be too low for microbiota analysis. The sample size was initially calculated for the 6-month time point, according to physiological data and fatty acid composition variability based on preliminary data. The 3-month time point was considered as an intermediate time point, whereas the 6-month time point was considered as the “main” end time-point to complete a previous study. Although some differences in microbiota profiles were observed among groups, previous studies have suggested a sample size >10 for sufficient power based on diversity metrics and microbiota composition [57,58]. Thus, the lower sample size of our study could result in undetected small effects. Moreover, the cage effect was not corrected for in the microbiota analysis, while co-caging also impacts microbiota composition [85]. However, the highly controlled environment of the animal facility, and the standardization of sample collection, storage and extraction, helped in minimizing the technical variation and bias that could increase variability within samples and reduce statistical power. Further investigations are, however, necessary in order to confirm this maternal LA-diet effect upon offspring microbiota composition. Another limitation of the study is the fact that we investigated only one section of the rat intestine, i.e., the cecum and proximal colon, while *n*-6 PUFA could have also affected more proximal regions of the gut. Indeed, Druart et al. observed lower yet detectable amounts of CLA in the ileum of mice gavaged with LA-rich oil, suggesting the possible effect of *n*-6 enriched diet on the small intestine microbiota [51]. Finally, while we observed significant effects of the maternal and weaning LA-diets on adipose tissue morphology and key lipid metabolism gene expression, we did not explore glucose homeostasis in our rats. A strong link between gut microbiota and glucose homeostasis has also been described [86], and would deserve investigations in this model.

## 5. Conclusions

We showed the differential and interactive effects of high LA intake in maternal and weaning diets on intestinal and adipose tissues, as well as on gut microbiota in young and older adult offspring. Despite a low n value, we observed that the gut microbiota of offspring at both 3 and 6 months were altered by the early perinatal exposure to LA, suggesting that high maternal LA intake may shape the gut microbiota of offspring and induce differential metabolic responses to diet exposure. As such, this study highlights the need for additional work to investigate how the increased consumption of LA in females may affect offspring outcomes. 

## Figures and Tables

**Figure 1 nutrients-12-03451-f001:**
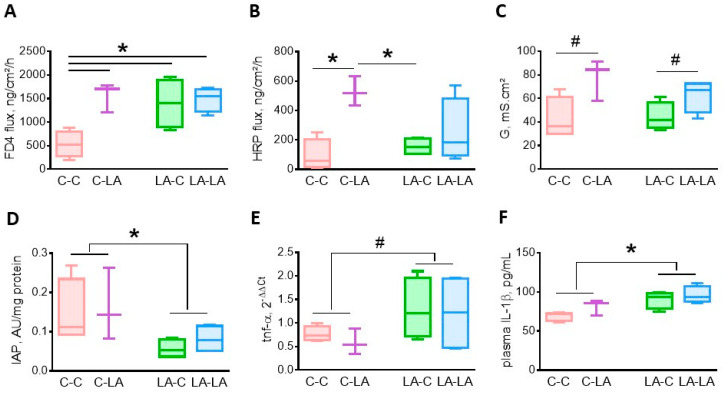
Impact of maternal and weaning LA-diets on large intestine barrier function and systemic inflammation at 3 months of age. Permeability to small (**A**) or large probes (**B**), tissue conductance (**C**) of the cecum, colonic intestinal alkaline phosphatase (IAP) activity (**D**) and tumor necrosis-α (*tnf-α)* mRNA levels (**E**) and plasma interleukin (IL)-1β concentrations (**F**) were determined in C-C, C-LA, LA-C and LA-LA rats at 3 months of age. Two-way ANOVA were performed to test the maternal, weaning and interaction effects. * *p* ≤ 0.05, ^#^
*p* ≤ 0.08, *n* = 4/group, except C-LA, where *n* = 3. LA: linoleic acid, C: control, FD4: FITC-dextran 4000, HRP: Horseradish Peroxidase, G: tissue conductance.

**Figure 2 nutrients-12-03451-f002:**
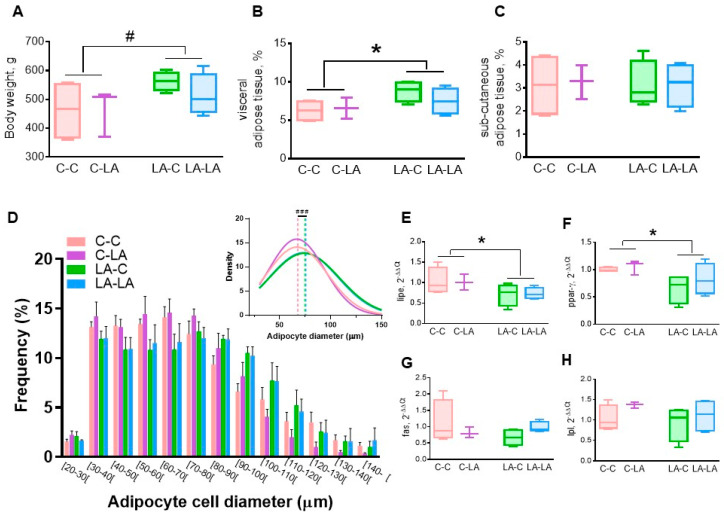
Impact of maternal and weaning LA-diets on adipose tissue physiology at 3 months of age. Body weight (**A**), visceral adiposity (**B**) and sub-cutaneous adiposity (**C**) were determined in CC, C-LA, LA-C and LA-LA rats at 3 months of age. Distribution of adipocyte diameters (**D**, insert represents the mean adipocyte diameter), hormone-sensitive lipase (*lipe*, **E**), peroxisome proliferator-activated receptor-γ (*ppar-γ*, **F**), fatty acid synthase (*fas*, **G**), and lipoprotein lipase (*lpl*, **H**) mRNA levels were determined in the epididymal adipose tissue. Two-way ANOVAs were performed to test the maternal, weaning and interaction effects. * *p* ≤ 0.05, ^#^
*p* ≤ 0.08, *n* = 4/group, except C-LA, where *n* = 3. LA: linoleic acid, C: control. ^###^: significant difference (*p* ≤ 0.05) between curve peaks.

**Figure 3 nutrients-12-03451-f003:**
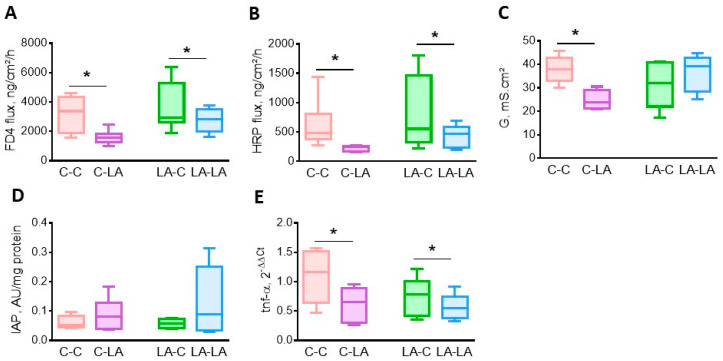
Impact of maternal and weaning LA-diets on large intestine barrier function at 6 months of age. Permeability to small (**A**) or large probes (**B**), tissue conductance (**C**) of the cecum, colonic intestinal alkaline phosphatase (IAP) activity (**D**) and tumor necrosis-α (*tnf-α)* mRNA levels (**E**) were determined in C-C, C-LA, LA-C and LA-LA rats at 6 months of age. Two-way ANOVAs were performed to test the maternal, weaning and interaction effects. * *p* ≤ 0.05, *n* = 6/group. LA: linoleic acid, C: control, FD4: FITC-dextran 4000, HRP: horseradish peroxidase, G: tissue conductance.

**Figure 4 nutrients-12-03451-f004:**
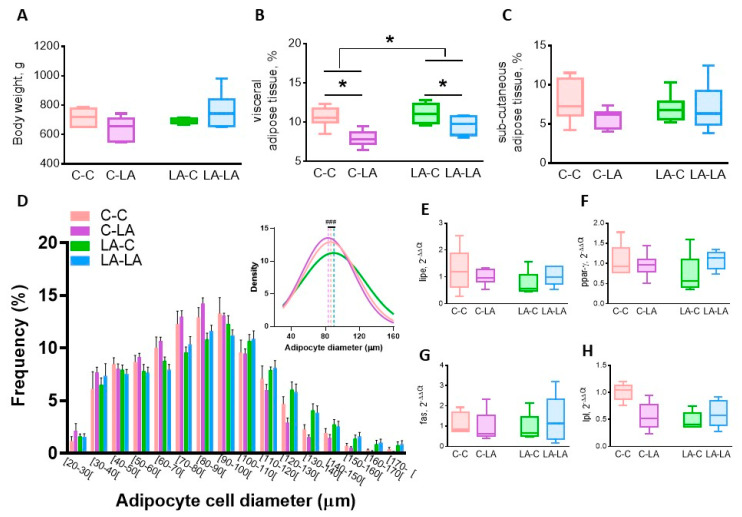
Impact of maternal and weaning LA-diets on adipose tissue physiology at 6 months of age. Body weight (**A**), visceral adiposity (**B**) and sub-cutaneous adiposity (**C**) were determined in C-C, C-LA, LA-C and LA-LA rats at 6 months of age. Distribution of adipocyte diameters (**D**, insert represents the mean adipocyte diameter), hormone-sensitive lipase (*lipe*, **E**), peroxisome proliferator-activated receptor-γ (*ppar-γ*, **F**), fatty acid synthase (*fas*, **G**), and lipoprotein lipase (*lpl*, **H**) mRNA levels were determined in the epididymal adipose tissue. Two-way ANOVAs were performed to test the maternal, weaning and interaction effects. * *p* ≤ 0.05, *n* = 6/group. LA: linoleic acid, C: control. ^###^: significant difference (*p* ≤ 0.05) between curve peaks.

**Figure 5 nutrients-12-03451-f005:**
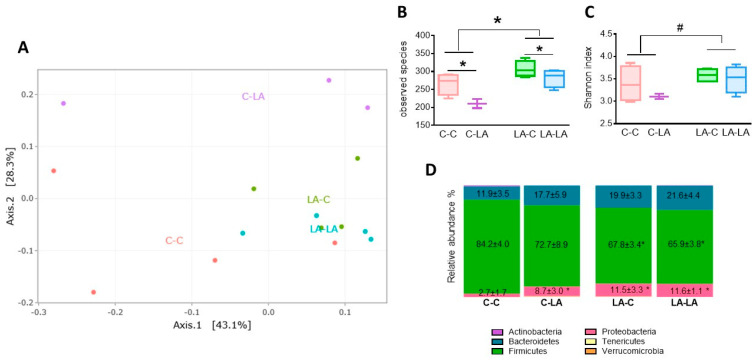
Impact of maternal and weaning LA-diets on microbiota composition at 3 months of age. Cecal microbiota composition of C-C, C-LA, LA-C and LA-LA rats was determined at 3 months of age through 16S rRNA gene sequencing. **A**. Principal coordinates analysis based on weighted Unifrac distances. **B**. Microbiota richness (observed operational taxonomic unit (OTU). **C**. Microbiota diversity estimated by the Shannon index. **D**. Main phyla relative abundance (mean ± SEM). Two-way ANOVAs were performed to test the maternal, weaning and interaction effects. Panels B–C: * indicates significant effect (*p* ≤ 0.05), ^#^ indicates a trend for significance (*p* ≤ 0.07), Panel D: * indicates maternal diet effect (*p* ≤ 0.05) *n* = 4/group, except C-LA, where *n* = 3. LA: linoleic acid, C: control, SEM: standard error of the mean.

**Figure 6 nutrients-12-03451-f006:**
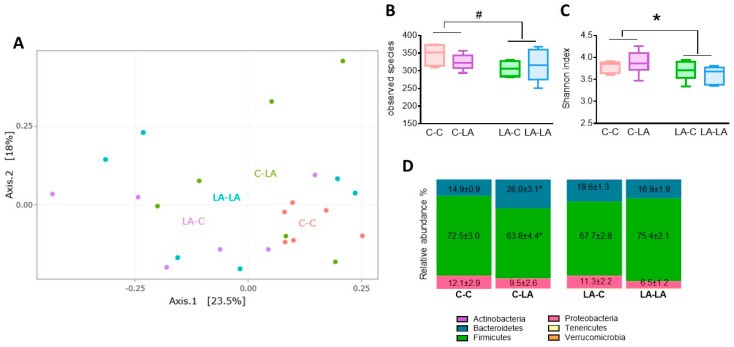
Impact of maternal and weaning LA-diets on microbiota composition at 6 months of age. The cecal microbiota composition of C-C, C-LA, LA-C and LA-LA rats was determined at 6 months of age through 16S rRNA gene sequencing. **A**. Principal coordinates analysis based on weighted Unifrac distances. **B**. Microbiota richness (observed operational taxonomic unit (OTU)). **C**. Microbiota diversity estimated by the Shannon index. **D**. Main phyla relative abundance (mean ± SEM). Two-way ANOVAs were performed to test the maternal, weaning and interaction effects. Panels B–C: letters indicate significant difference (*p* ≤ 0.05), ^#^
*p* ≤ 0.07, Panel D: * indicates difference between C-C and C-LA (*p* ≤ 0.05), *n* = 6/group. LA: linoleic acid, C: control, SEM: standard error of the mean.

**Figure 7 nutrients-12-03451-f007:**
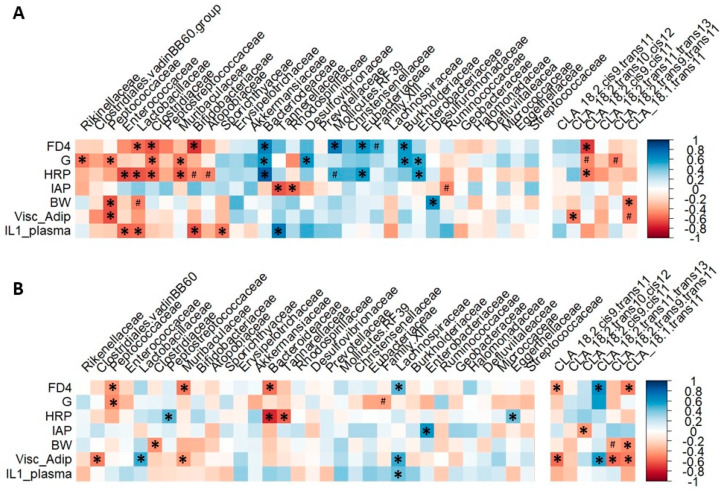
Correlations between family relative abundances and conjugated linoleic acid (CLA)s, and rat phenotype key parameters. Spearman rank correlations were performed at 3 months (**A**) and 6 months (**B**) of age between family relative abundances or CLAs and gut barrier, adiposity and systemic inflammation parameters, and represented as a correlation matrix between parameters. * *p* < 0.05, ^#^
*p* < 0.06. FD4: FITC-dextran 4000, G: tissue conductance, HRP: horseradish peroxidase, IAP: intestinal alkaline phosphatase, Visc_Adip: visceral adiposity, BW: body weight.

**Table 1 nutrients-12-03451-t001:** Fatty acid composition of offspring epididymal adipose tissue at 3 months of age.

Maternal Diet	C		LA		*p*-Value		
Weaning Diet	C	LA	C	LA	Maternal Diet	Weaning Diet	x ^1^
SFA	24.5 (0.7)	25.9 (1.3)	25.5 (0.4)	24.6 (1.3)	0.89	0.76	0.27
MUFA	67.5 (0.5)	31.2 (1.1)	66.4 (0.2)	30.7 (1.6)	0.44	<0.001	0.79
*n*-6 PUFA	6.4 (0.3)	41.6 (1.8)	6.7 (0.2)	43.2 (1.4)	0.39	<0.001	0.54
*18:2 n-6*	6.2 (0.2)	39.8 (2.0)	6.5 (0.2)	41.3 (1.4)	0.44	<0.001	0.59
*20:4 n-6*	0.1 (0.0)	0.8 (0.1)	0.1 (0.0)	0.9 (0.1)	0.33	<0.001	0.50
*n*-3 PUFA	1.6 (0.0) ab	1.6 (0.0) ab	1.4 (0.1) a	1.8 (0.0) b	0.95	0.03	0.01
*n*-6/*n*-3	4.0 (0.1) a	26.7 (0.6) b	4.9 (0.5) a	24.2 (0.3) c	0.07	<0.001	<0.01

^1^ maternal diet × weaning diet interaction. Two-way ANOVAs were performed to test the maternal, weaning and interaction effects. a,b,c: *p* < 0.05, *n* = 4/group, except C-LA, where *n* = 3. C: control, LA: linoleic acid, SFA: saturated fatty acids, MUFA: mono-unsaturated fatty acids, PUFA: poly-unsaturated fatty acids.

**Table 2 nutrients-12-03451-t002:** Hepatic conjugated linoleic acid concentrations at 3 months of age.

Maternal Diet	C		LA		*p*-Value		
Weaning Diet	C	LA	C	LA	Maternal Diet	Weaning Diet	x ^1^
18:2 *cis*-9, *trans*-11	0.012 (0.003) a	0.014 (0.002) a	0.010 (0.002) a	0.024 (0.002) b	0.08	0.1	0.02
18:2 *trans*-10, *cis*-12	0.008 (0.002)	0.057 (0.03)	0.006 (0.001)	0.013 (0.005)	0.08	0.04	0.11
18:2 *cis*-9, *cis*-11	0.026 (0.009)	0.012 (0.003)	0.039 (0.006)	0.013 (0.004)	0.30	0.01	0.40
18:2 *trans*-11, *trans*-13	0.011 (0.003)	0.009 (0.002)	0.006 (0.001)	0.007 (0.004)	0.35	0.80	0.61
18:2 *trans*-9, *trans*-11	0.011 (0.003)	0.004 (0.009)	0.008 (0.003)	0.015 (0.005)	0.30	0.87	0.08
18:1 *trans*-11	0.042 (0.009) ab	0.031 (0.005) ab	0.027 (0.005) a	0.050 (0.004) b	0.77	0.34	0.02

^1^ maternal diet × weaning diet interaction, Two-way ANOVAs were performed to test the maternal, weaning and interaction effects. a,b: *p* < 0.05, *n* = 4/group, except C-LA, where *n* = 3. C: control, LA: linoleic acid.

**Table 3 nutrients-12-03451-t003:** Fatty acid composition of offspring epididymal adipose tissue at 6 months of age.

Maternal Diet	C		LA		*p*-Value		
Weaning Diet	C	LA	C	LA	Maternal Diet	Weaning Diet	x ^1^
SFA	21.7 (0.3)	23.1 (0.3)	22.9 (0.4)	24.2 (0.9)	0.05	0.02	0.96
MUFA	70.8 (0.3)	30.4 (0.7)	69.2 (0.3)	29.8 (0.7)	0.08	<0.001	0.36
*n*-6 PUFA	6.4 (0.1)	44.9 (0.9)	6.9 (0.1)	44.5 (1.4)	0.99	<0.001	0.59
*18:2 n-6*	6.1 (0.1)	43.4 (0.8)	6.5 (0.1)	42.8 (1.5)	0.94	<0.001	0.60
*20:4 n-6*	0.2 (0.0)	0.7 (0.1)	0.1 (0.0)	0.8 (0.1)	0.27	<0.001	0.50
*n*-3 PUFA	1.0 (0.0)	1.3 (0.1)	1.0 (0.0)	1.3 (0.0)	0.08	0.02	0.66
*n*-6/*n*-3	6.6 (0.1)	34.9 (0.8)	6.6 (0.0)	33.2 (1.4)	0.31	<0.001	0.31

^1^ maternal diet × weaning diet interaction. Two-way ANOVAs were performed to test the maternal, weaning and interaction effects. *n* = 6/group, C: control, LA: linoleic acid, SFA: saturated fatty acids, MUFA: mono-unsaturated fatty acids, PUFA: poly-unsaturated fatty acids.

**Table 4 nutrients-12-03451-t004:** Hepatic conjugated linoleic acid concentrations at 6 months of age.

Maternal Diet	C		LA		*p*-Value		
Weaning Diet	C	LA	C	LA	Maternal Diet	Weaning Diet	x ^1^
18:2 *cis*-9, *trans*-11	0.015 (0.001)	0.033 (0.004)	0.011 (0.002)	0.029 (0.002)	0.21	<0.001	0.98
18:2 *trans*-10, *cis*-12	0.025 (0.013)	0.016 (0.003)	0.020 (0.007)	0.016 (0.004)	0.75	0.37	0.78
18:2 *cis*-9, *cis*-11	0.033 (0.004)	0.013 (0.002)	0.024 (0.006)	0.014 (0.004)	0.34	<0.01	0.25
18:2 *trans*-11, *trans*-13	0.013 (0.003)	0.006 (0.001)	0.012 (0.002)	0.010 (0.003)	0.64	0.04	0.35
18:2 *trans*-9, *trans*-11	0.022 (0.009)	0.026 (0.004)	0.008 (0.001)	0.010 (0.003)	<0.01	0.54	0.89
18:1 *trans*-11	0.030 (0.005)	0.045 (0.006)	0.016 (0.004)	0.037 (0.01)	0.13	0.02	0.64

^1^ maternal diet × weaning diet interaction. Two-way ANOVAs were performed to test the maternal, weaning and interaction effects. *n* = 6/group. C: control, LA: linoleic acid.

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
