# Peer review of "Maternal Linoleic Acid Overconsumption Alters Offspring Gut and Adipose Tissue Homeostasis in Young but Not Older Adult Rats"

_nutrients, 2020, doi:10.3390/nu12113451_

Round 1

Reviewer 1 Report

Given the persistent high consumption of linoleic acid in human diets, the question of what effect linoleic acid has on the body is major and highly relevant. Marchix and coauthors address this question at several levels, the effect on body fat composition, on microbiome composition, on gut permeability, on inflammatory markers, body weight and body fat amount, and on adipocytes. They analyze these physiologies in rat mothers, weaned pups, and pups at 3 and 6 months of age through use an ingenious study design: mothers on a control or linoleic acid enriched diet with their pups weaned onto either the same or opposite diet. While the experimental design and intent of their study are strong, the number of animals studied in each group hampers the power and rigor of their work, especially regarding microbiome analysis.

Major concerns

The sample size is too low for microbiome analysis. Likely your study requires a minimum of 10 animals per group for sufficient power. Built into this concern is the cage effect. How many animals were co-caged together post weaning? Cage (both birth and weaned cage) should also be incorporated into the statistical analysis. As well, there is concern over the reproducibility of the microbiome: if you repeated the experiment, would you observe the same OTU shifts? Please consult PMID 25819674, PMID 27760558.

For the group of animals sacrificed at weaning, dams and pups (n=3), how many were dams vs pups?

Minor comments

Figure S1B: In the boxes at the bottom, “tissu” should be “tissue”

Page 5 line 204: extra space after composition

Text at p5 line 220 says p=0.07 and the legend for Fig 1 says trending is p<0.06.

More typical to say “weighted Unifrac”

Figure 3: Reorder 3F-> 3E, 3H -> 3F, 3E->3G, 3G-> 3H

With enough N, Fig S2 is very interesting and could be brought to the main text

Page 13, line 64: “duet” -> “diet”

Page 16, line 7: space after Lactobacillaceae

Author Response

The authors thank the reviewer for his/her constructive and thoughtful comments.

Given the persistent high consumption of linoleic acid in human diets, the question of what effect linoleic acid has on the body is major and highly relevant. Marchix and coauthors address this question at several levels, the effect on body fat composition, on microbiome composition, on gut permeability, on inflammatory markers, body weight and body fat amount, and on adipocytes. They analyze these physiologies in rat mothers, weaned pups, and pups at 3 and 6 months of age through use an ingenious study design: mothers on a control or linoleic acid enriched diet with their pups weaned onto either the same or opposite diet. While the experimental design and intent of their study are strong, the number of animals studied in each group hampers the power and rigor of their work, especially regarding microbiome analysis.

Major concerns

The sample size is too low for microbiome analysis. Likely your study requires a minimum of 10 animals per group for sufficient power. Built into this concern is the cage effect. How many animals were co-caged together post weaning? Cage (both birth and weaned cage) should also be incorporated into the statistical analysis. As well, there is concern over the reproducibility of the microbiome: if you repeated the experiment, would you observe the same OTU shifts? Please consult PMID 25819674, PMID 27760558.

We agree with the reviewer that the sample size may be too low for microbiota analysis. Regarding the experimental design, the 3-month time point was initially considered as an intermediate time point whereas the 6-month time point was considered as the “main” end-time point to complete a previous study. The sample size at 6-month was calculated based on a power analysis in R according to other measurements including physiological data and fatty acid composition variability based on preliminary data. However, we did not have preliminary data on microbiota composition in this particular dietary context to perform a power analysis. The n number for the 3-month time point was not calculated according to this power analysis. We did not anticipate such interesting and contradicting results within 3- and 6-month time points.

As pointed by the reviewer, reproducibility of the results in science is a concern, especially for microbiota studies. Animal facility environment, cage effects and batches of animals are among the major determinants of microbiota composition, interfering with data reproducibility from one lab to the other. Thus, we believe that the highly controlled environment of the animal facility and the “standardization” of sample collection, storage and extraction helped minimizing technical variation and bias that could increase variability within samples and reduce statistical power. Also, the number of samples may be too low to achieve statistical power which could result in failing to detect real but small effects (doi.org/10.1053/j.gastro.2019.11.305). Repeating the experiment and increasing the sample size will probably identify same bacteria community changes but could allow us to unravel more specific microbiota changes. However, OTU shifts cannot be excluded regarding the multiple confounding factors that regulates microbiota composition. As suggested by the reviewer, our study provides some interesting preliminary data on how enriched LA diet can modify gut microbiota in offspring but further experiments are required to complete and specify the impact of high LA intake on offspring gut microbiota over time. We added a section in the discussion on the limitations of the study, which includes this point (lines 163-74).

For the group of animals sacrificed at weaning, dams and pups (n=3), how many were dams vs pups?

Three dams / dietary group and 3 pups / dietary dam  group were studied for tissue fatty acid composition. This is now clearly specified in the M&M section (line 125) and in Supplementary figure 1B.

Minor comments

Figure S1B: In the boxes at the bottom, “tissu” should be “tissue”

corrected

Page 5 line 204: extra space after composition

corrected

Text at p5 line 220 says p=0.07 and the legend for Fig 1 says trending is p<0.06.

We considered trending at P≤0.07. This has been corrected in the legend.

More typical to say “weighted Unifrac”

Corrected throughout the manuscript

Figure 3: Reorder 3F-> 3E, 3H -> 3F, 3E->3G, 3G-> 3H

This has been changed accordingly.

With enough N, Fig S2 is very interesting and could be brought to the main text

Unfortunately, the N number cannot be increased and we decided to keep Figure S2 in the supplementary data.

Page 13, line 64: “duet” -> “diet”

Corrected

Page 16, line 7: space after Lactobacillaceae

Corrected

Reviewer 2 Report

Marchix er al. investigate the effects of LA intake in maternal diet and after weaning in young and adult rats on gut microbiota, intestinal permeability and adipose tissue histology. The work is of potential interest to a wide readership but offer a few limitations, which need to be addressed. 

major:

  • since this is a study focusing among other on the effects of diet on microbiome changes, please extend the methods section on this subject accordingly both for experimental and analyses sections. This is important because some of the described methods in earlier works, can not be applicable to other sequencing runs. 
  • the 16S rRNA sequences should be uploaded to open access repositories and code for analysis made available. 
  • crucial data is missing such as how the rats were caged and what methods the authors used to regress variables on bacterial composition (Ordistep i.e. ? one is left wondering).  
  • the way the data is presented in several figures is not informative and statistics are partly missing in the text, where they are needed to support the figures. Figures have low resolution partly ( this is minor) and need to be corrected. 
  • The authors report differential abundances in OTUs but present data as high ranked as family level, which is hard to understand. Were there no relevant results at genus level? 

Minors:

  1. introduction:
    • line 41: change "might be involved" to are related/linked to greater risk. 
    • line 60: add reference after "behavior"
    • line 61: add the appropriate reference after energy harvest, i.e.:Bäckhed F., Ding H., Wang T., Hooper L.V., Koh G.Y., Nagy A., Semenkovich C.F., Gordon J.I. The gut microbiota as an environmental factor that regulates fat storage.
    • line 62: add "altered" before production of bacterial metabolites (SCFAs are suggested to be detrimental by the sentence currently there, which is not accurate).
    • line 64: the microbiota "is established" and not "establishes" (it is primarily not an active process and your data supports this by showing that diet influences microbiota configuration).
    • same for line 65
    • omit comma after microbiota in line 69
    • add comma before which and before where in lines 74 and lines 79, replace "couples of" with "few" in line 79
    • lines 79-84: you suggest that the role of n-6-PUFA on intestinal microbiota isn't completely elucidated right after you describe a study where transgenic mice endogenously produce n-6-PUFA independent of diet and their direct effects on microbiota are studied. This is somewhat of a contradiction. You might want to specify that the overreaching effect of maternal dietary n-6PUFA intake in the light of moderate fat diet with differing proportions of n-6 PUFA after weaning. the reference 28 itself is ok, but glancing over the paper I was shocked by the amount of mistakes related to bacterial nomenclature. If you are able to find a better paper, please replace that reference. 
    • line 85: alter instead of "alters" (does not significantly alter).
    • line 88: replace "role upon" with "impact on"
    • lines 92-96: Please omit completely. these are results and already described in the abstract. 
  2. Materials and methods:
    • please clarify the subsections as such: for example in line 102 and 111, you write: experimental diets. This is obviously a subheading. correct punctuation accordingly. 
    • be attentive to superfluous spaces: e.g. line 106, line 204 to name a few. Run the paper through a spelling software. 
    • experimental design: please add the exact number of n in each group within the S figure. which group had 4 and which had 5 and why were they unbalanced, why did you use n=4 for 3 months and n=6 for 6 months and why not balanced? 5 and 5? how were the animals caged? this is crucial for microbiome surveys. 
    • line 120: replace "for" with from.
    • line 135: it is the 16S rRNA gene not 16S rDNA.. 
    • line 133: the title is erroneous. There you do not describe micobiota analysis but amplification of V3-V4 of 16S rRNA gene, you introduce sequencing and that is it. Considering at least of the results are about microbiota changes, please extend the section to include microbiota analyses. 
    • line 141 and 144: why are reference numbers jumping all the way to 60? please correct. 
    • lines 142, line 175: you have a single sentence in each of these sections. Please join them into one section with other shorter sections. 
    • real-time PCR: specify R2
    • line 168: replace "meet" with met. 
    • statistical analyses: add a section of qPCR data analysis as well as metagenomic data analyses. Cite R. What packages did you use when you use R. Please specify the correlation method (Pearson? Spearman?).
  3. Results:
    • In general in results: Please specify n each figure and result section the sample size used for the test. 
    • lines 194-197: this belongs in the methods section. 
    • line 206: are the statistics shown for both tissues at once? these numbers can't be found in the table, also the result you report is insignificant but your table shows significant results 
    • lines 214: what was used to regress variables onto bacterial composition? 
    • Figure 1A: the resolution of the figure is suboptimal, please export in more high resolution if possible. 
    • Figure 1 B-J: it is misleading to represent this data this way. Best is to report boxplots with median and whiskers for the readers to really see what it significant or not (considering you have 4-6 individual results, this should still be ok). Also please add in the legend the test that was used. The legend on top doesn't help clarify the figure and the plots are just not informative. 
    • Figure D: it seems to me that LA-C and C-C have similar amounts of proteobacteria. Can you provide log2foldchange or foldchange values to what you report? p-values alone are not enough. 
    • it is reported that no effects of weaning diet is observed but just optically the differences in the weaning diets seem significant compared to what is reported on maternal diet. Otherwise could there be a labelling issue? 
    • Line 258: Figure 2A, Again boxplots might be more informative unless you have very few observations. SEM and means don't convey enough information to get any feeling for the data. 
    • Table 1 is broken down on 2 pages and this makes it hard to read. please correct.
    • Figure 3: here, means and SEM still make it possible to get a slight feeling for the data. Still I believe it is not the best way to convey data. Please correct according to the suggestion above.
    • Figure 3D: the resolution of the distribution graph is low. One can not make out the y-axis title on a printed paper (assuming it is density). Also the gray is very light and I have no idea if at the top between the medians it is an asterisk or letters that one sees. Please make it slightly larger and optimize the resolution.
    • please introduce abbreviations. CLA is mentioned already at line 94. Spell it out the first time. 
    • line 313: as a reflection (not reflect)
    • Starting with section 3.3: are these supplementary results?? it seems not as you have main figures in there. The line numbers start from 1 again in this section. This is an unfortunate formatting error.  So from now on the lines I am referring to, relate to the section 3.3
    • line 47 : introduce HRP
    • again 3.4 section starts with line 1 , what is the point of having line numbers if a reviewer can't use them.. 
    • it is called Spearman rank correlation not mathematical Spearman corr.. 
    • were all families correlated or only specific ones selected?
    • also why is the work only at family level, are there no relevant genera showing up anywhere? 
  4. Discussion: LA born rats are less impacted by change of diet in weaning, they display lower amounts of Lactobacillales and Clostridiaceae. And while these families are negatively associated with gut permeability, the richness in the LA born rats seemed higher initially. Richness on the other hand is usually a hallmark of a healthier microbiome. how do the authors relate their results to the current literature and how do they explain the initially higher richness in LA born rats?
    • line 76: replace "on the opposite" with "on the other hand". 
    • line 82: add comma before while. Generally go through punctuation with grammarly (could be taken care of by the journal). 
    • line 86: couples of study is not an expression. please correct. 
    • line 92: delete the extra dot. same line: some genera not some genus. 
    • line 98: replace participate with contribute
    • line 121: add comma before while
    • line 128: evidence (singular not plural)
    • at 6 months of age LA weaning diet fed rats exhibit a healthier phenotype. This practically contradicts the authors' observations and discussion on the connection between microbiome, permeability and adiposity. More so, since the microbiota in LA born and weaned rats is described as relatively stable (except C-LA). How do the authors explain this?
    • please add a section on limitations. this might include the low power (n smaller than 6 in each group is tricky) and any other shortcomings that the authors weren't able to address (adiposity is nice but it is not the most relevant metabolic readout). 

Author Response

The authors thank the reviewer for his/her constructive and thoughtful comments.

Marchix er al. investigate the effects of LA intake in maternal diet and after weaning in young and adult rats on gut microbiota, intestinal permeability and adipose tissue histology. The work is of potential interest to a wide readership but offer a few limitations, which need to be addressed. 

Major:

  • since this is a study focusing among other on the effects of diet on microbiome changes, please extend the methods section on this subject accordingly both for experimental and analyses sections. This is important because some of the described methods in earlier works, can not be applicable to other sequencing runs. 

The methods used for microbiota analysis, including bioinformatical and statistical analysis, are now thoroughly described in the M&M section (lines 140-178).

  • the 16S rRNA sequences should be uploaded to open access repositories and code for analysis made available.

Data have been uploaded and details for downloading are provided in the manuscript.

  • crucial data is missing such as how the rats were caged and what methods the authors used to regress variables on bacterial composition (Ordistep i.e. ? one is left wondering).  

Description of rat housing and mathematical methods is now provided in the M&M section (lines 122-123).

  • the way the data is presented in several figures is not informative and statistics are partly missing in the text, where they are needed to support the figures. Figures have low resolution partly ( this is minor) and need to be corrected. 

Figures have been changed for better clarity of the statistical differences.

  • The authors report differential abundances in OTUs but present data as high ranked as family level, which is hard to understand. Were there no relevant results at genus level? 

Two analyses were performed:

  • one at the family level, using ANOVA and presented either as plots in Figures 1 and 4 for the families exhibiting the more significant changes and in Tables S4 and S7 for all the families, including those which do not vary
  • one at the OTU level, using Deseq2 analysis comparing the weaning diet effect within maternal diet group. Data are presented in Tables S5 and S6 and Tables S7 and S8

Deseq2 is the preferred method when working at the OTU level since it takes into account the omic structure of the OTU table (sparse OTUs, i.e. many samples may not contain a given taxon). However, Deseq2 analysis can only performed comparison between 2 groups, which was not relevant with our experimental design with 2 factors (maternal and weaning diet).  We therefore decided to use classical statistical methods and work at the family level (not subjected to the omic structure limitations than the OTU table) to highlight possible interactions between dietary factors.

Minors:

  1. introduction:
    • line 41: change "might be involved" to are related/linked to greater risk. 

corrected

    • line 60: add reference after "behavior"

done

    • line 61: add the appropriate reference after energy harvest, i.e.:Bäckhed F., Ding H., Wang T., Hooper L.V., Koh G.Y., Nagy A., Semenkovich C.F., Gordon J.I. The gut microbiota as an environmental factor that regulates fat storage.

changed

    • line 62: add "altered" before production of bacterial metabolites (SCFAs are suggested to be detrimental by the sentence currently there, which is not accurate).

added

    • line 64: the microbiota "is established" and not "establishes" (it is primarily not an active process and your data supports this by showing that diet influences microbiota configuration).

corrected

    • same for line 65

corrected

    • omit comma after microbiota in line 69

done

    • add comma before which and before where in lines 74 and lines 79, replace "couples of" with "few" in line 79

corrected

    • lines 79-84: you suggest that the role of n-6-PUFA on intestinal microbiota isn't completely elucidated right after you describe a study where transgenic mice endogenously produce n-6-PUFA independent of diet and their direct effects on microbiota are studied. This is somewhat of a contradiction. You might want to specify that the overreaching effect of maternal dietary n-6PUFA intake in the light of moderate fat diet with differing proportions of n-6 PUFA after weaning. the reference 28 itself is ok, but glancing over the paper I was shocked by the amount of mistakes related to bacterial nomenclature. If you are able to find a better paper, please replace that reference. 

Studies with fat-1 and fat-2 mice gave very interesting results on the impact of n-6/n-3 PUFA animal status on gut microbiota. However, these genetic models do not give information on dietary n-6 PUFA consumption and one could imagine that the effect would be different from that of n-6 PUFA tissue status. That’s why we believe that the impact of dietary n-6 PUFA is not well understood. We added a sentence (lines 80-82) in the introduction. We did not find another reference with Fat-2 mice and decide to keep this one.

    • line 85: alter instead of "alters" (does not significantly alter).

corrected

    • line 88: replace "role upon" with "impact on"

done

    • lines 92-96: Please omit completely. these are results and already described in the abstract. 

The sentences were deleted

  1. Materials and methods:
    • please clarify the subsections as such: for example in line 102 and 111, you write: experimental diets. This is obviously a subheading. correct punctuation accordingly. 

We deleted the experimental diet and experiment design sentences.

    • be attentive to superfluous spaces: e.g. line 106, line 204 to name a few. Run the paper through a spelling software. 

We carefully checked for double spaces throughout the manuscript.

    • experimental design: please add the exact number of n in each group within the S figure. which group had 4 and which had 5 and why were they unbalanced, why did you use n=4 for 3 months and n=6 for 6 months and why not balanced? 5 and 5? how were the animals caged? this is crucial for microbiome surveys. 

We specified the number of animals studied per dietary group in supplementary figure 1B and for each figure / table. Within a dietary group rats were housed in 5 different cages with 2 rats/cage. This is now clearly described in the M&M section (lines 122-123). Regarding the experimental design, the 3-month time point was initially considered as an intermediate time point where as the 6-month time point was considered as the “main” end-time point to complete a previous study (and thus the n was increased for this time point). We did not anticipate such interesting and contradicting results within 3- and 6-month time points. Further analysis will be required to investigate the two time periods.

    • line 120: replace "for" with from.

replaced

    • line 135: it is the 16S rRNA gene not 16S rDNA.. 

corrected

    • line 133: the title is erroneous. There you do not describe micobiota analysis but amplification of V3-V4 of 16S rRNA gene, you introduce sequencing and that is it. Considering at least of the results are about microbiota changes, please extend the section to include microbiota analyses. 

We now fully describe the sequencing, bio-informatics and statistical steps in this section (line 139-178).

    • line 141 and 144: why are reference numbers jumping all the way to 60? please correct. 

References have been checked and corrected when necessary.

    • lines 142, line 175: you have a single sentence in each of these sections. Please join them into one section with other shorter sections. 

These two short sections have been combined with other.

    • real-time PCR: specify R2

The R2 has been specified in the M&M sections (lines 197-199).

    • line 168: replace "meet" with met. 

done

    • statistical analyses: add a section of qPCR data analysis as well as metagenomic data analyses. Cite R. What packages did you use when you use R. Please specify the correlation method (Pearson? Spearman?).

The statistical method section has been changed accordingly (lines 230-237).

  1. Results:
    • In general in results: Please specify n each figure and result section the sample size used for the test. 

The n number was added in the legend of each figures and tables.

    • lines 194-197: this belongs in the methods section. 

The sentences were deleted.

    • line 206: are the statistics shown for both tissues at once? these numbers can't be found in the table, also the result you report is insignificant but your table shows significant results 

Statistic data at the end of previous line 206 (now line 252) refered to pup body weight, which was not significantly different between the two maternal diet groups. Table S2 shows statistics for individual tissues.

    • lines 214: what was used to regress variables onto bacterial composition? 

The adonis function of the phyloseq R package was used to detect significant differences between the dietary groups or between maternal dietary groups or weaning dietary groups.

    • Figure 1A: the resolution of the figure is suboptimal, please export in more high resolution if possible. 

done

    • Figure 1 B-J: it is misleading to represent this data this way. Best is to report boxplots with median and whiskers for the readers to really see what it significant or not (considering you have 4-6 individual results, this should still be ok). Also please add in the legend the test that was used. The legend on top doesn't help clarify the figure and the plots are just not informative. 

Figures were changed to box and whisker plots, statistical tests used were added in the legend and signs to show statistical differences were changed.

    • Figure D: it seems to me that LA-C and C-C have similar amounts of proteobacteria. Can you provide log2foldchange or foldchange values to what you report? p-values alone are not enough. 

We added the mean relative abundance for the 3 major phyla (Firmicutes, Bacteroidetes, Proteobacteria) on the plots. The asterisk in Figure 1D indicates a significant overall maternal effect. This is specified in the legend.

    • it is reported that no effects of weaning diet is observed but just optically the differences in the weaning diets seem significant compared to what is reported on maternal diet. Otherwise could there be a labelling issue? 

We hope the addition of mean value for the phyla will help clarify this issue.

    • Line 258: Figure 2A, Again boxplots might be more informative unless you have very few observations. SEM and means don't convey enough information to get any feeling for the data. 

This has been changed accordingly.

    • Table 1 is broken down on 2 pages and this makes it hard to read. please correct.

This has been fixed.

    • Figure 3: here, means and SEM still make it possible to get a slight feeling for the data. Still I believe it is not the best way to convey data. Please correct according to the suggestion above.

This has been fixed.

    • Figure 3D: the resolution of the distribution graph is low. One can not make out the y-axis title on a printed paper (assuming it is density). Also the gray is very light and I have no idea if at the top between the medians it is an asterisk or letters that one sees. Please make it slightly larger and optimize the resolution.

The distribution graphs have been modified accordingly.

    • please introduce abbreviations. CLA is mentioned already at line 94. Spell it out the first time. 

This has been added.

    • line 313: as a reflection (not reflect)

corrected

    • Starting with section 3.3: are these supplementary results?? it seems not as you have main figures in there. The line numbers start from 1 again in this section. This is an unfortunate formatting error.  So from now on the lines I am referring to, relate to the section 3.3

Line numbers were added by the journal. I will have to check with them about this formatting error.

    • line 47 : introduce HRP

HRP and FD-4 are now introduced in the M&M section (lines 187-188).

    • again 3.4 section starts with line 1 , what is the point of having line numbers if a reviewer can't use them.. 

Similarly, I have to check with the journal.

    • it is called Spearman rank correlation not mathematical Spearman corr.. 

corrected

    • were all families correlated or only specific ones selected?

All the families were correlated to phenotypic data.

    • also why is the work only at family level, are there no relevant genera showing up anywhere? 

The structure of OTU tables (with sparse OTUs, i.e. many samples may not contain a given taxon) makes it difficult to perform robust mathematical correlations with phenotypic data. A variable reduction approach (such as WGCNA, which clusters the OTUs that co-variate) could have been performed. However, this would have given clusters with species from different taxa. We believe that working at the family level enables robust correlations and gives useful information as whether dietary n-6 PUFA impact specific families abundances, which is clearly the case in our study.

  1. Discussion: LA born rats are less impacted by change of diet in weaning, they display lower amounts of Lactobacillales and Clostridiaceae. And while these families are negatively associated with gut permeability, the richness in the LA born rats seemed higher initially. Richness on the other hand is usually a hallmark of a healthier microbiome. how do the authors relate their results to the current literature and how do they explain the initially higher richness in LA born rats?

We agree with the reviewer that this discrepancy between greater microbiota richness and a less healthy phenotype in 3-month old LA dam-born rats is intriguing. Reduced diversity is one of the parameters defining dysbiosis (DOI: 10.1111/cmi.12308). Gut microbiota richness is often positively associated to a healthy phenotype in adults. However, this might not be the case in younger individuals. For example, breast-fed infants display lower fecal microbiota richness than formula-fed infants (for review, DOI: 10.1007/s10620-020-06092-x), yet breast-feeding is associated with lower risk of developing disease. Our comparison of 3- and 6-month microbiota data (Figure S2) shows that the gut microbiota richness of rats born to C-mothers increases between 3 and 6-months of age, suggesting that 3-month old rats are still in a period of microbiota maturation. On the other hand, rats born to LA mothers had already a ‘mature’ microbiota in term of richness at 3 months of age. One can speculate that a precocious mature microbiota might not match host physiological and metabolism maturational stage. We added a sentence in the discussion (lines 57-60 of the discussion section). Lastly, the reason with richness was higher in LA born rats is unknown and further work will be needed to understand this result.

    • line 76: replace "on the opposite" with "on the other hand". 

done

    • line 82: add comma before while. Generally go through punctuation with grammarly (could be taken care of by the journal). 

done

    • line 86: couples of study is not an expression. please correct. 

We changed to ‘several studies’.

    • line 92: delete the extra dot. same line: some genera not some genus. 

corrected

    • line 98: replace participate with contribute

done

    • line 121: add comma before while

done

    • line 128: evidence (singular not plural)

corrected

    • at 6 months of age LA weaning diet fed rats exhibit a healthier phenotype. This practically contradicts the authors' observations and discussion on the connection between microbiome, permeability and adiposity. More so, since the microbiota in LA born and weaned rats is described as relatively stable (except C-LA). How do the authors explain this?

We agree with the reviewer that these results are counterintuitive and we had great difficulties to interpret these results. We believe that the rat phenotype at 6 months of age was strongly driven by n-6 PUFA derivatives such as CLA, OXLAM and eicosanoids linked to chronic weaning n-6 PUFA consumption. Indeed, we observed reduced visceral adiposity, greater steatosis as well as weaning diet-induced increase in these derivatives in this model (doi: 10.1016/j.jnutbio.2019.108241). Thus, one can speculate that the effects of these bioactive n-6 PUFA derivatives overcame a potential effect of the microbiota on host physiology.

    • please add a section on limitations. this might include the low power (n smaller than 6 in each group is tricky) and any other shortcomings that the authors weren't able to address (adiposity is nice but it is not the most relevant metabolic readout). 

A limitation section was added at the end of the discussion (lines 163-174 of the discussion section).

Reviewer 3 Report

I read the study by Marchix et al with great interest. While the effects of maternal diets are known to have an effect on the gut microbiome of their offspring showing this is associated specifically with LA on obesity is quite and important novel finding. I thought the study was well controlled. There are a few improvements that should be made.

1) what is the fat composition of the low LA diet? 

2) Please improve all the figures:  have dot plots or box and whisker plots with statistical significance shown in a more conventional manner.  As well, all the text is fuzzy and detracts from the high quality study.

Author Response

The authors thank the reviewer for his/her constructive and thoughtful comments.

I read the study by Marchix et al with great interest. While the effects of maternal diets are known to have an effect on the gut microbiome of their offspring showing this is associated specifically with LA on obesity is quite and important novel finding. I thought the study was well controlled. There are a few improvements that should be made.

1) what is the fat composition of the low LA diet? 

Both diets (low LA = C-diet and high LA =LA-­diet) were isolipidic (10% of fat, representing 21% of total energy). Fatty acid composition of both diets is presented in Figure S1A. Further details on the diets can be found in Marchix et al., J Nutr Biochem. 2020;75:10824.

2) Please improve all the figures:  have dot plots or box and whisker plots with statistical significance shown in a more conventional manner.  As well, all the text is fuzzy and detracts from the high quality study.

Figures have been changed to box and whisker plots and statistical difference clearly indicated as requested.

Round 2

Reviewer 1 Report

The problem of the underpowered microbiome study remains.

For this manuscript to be usable in the literature and to service naïve readers not familiar with microbiome study design, the limitations of the study need to be more upfront.  Especially since some readers do not read the conclusion, I suggest the authors make a note of the lack of sampling in the results section before any microbiota data is presented. Also all plots showing taxa should be moved to the supplement. This includes Figures 1D-J and Figures 4D-J.

Author Response

The problem of the underpowered microbiome study remains.

For this manuscript to be usable in the literature and to service naïve readers not familiar with microbiome study design, the limitations of the study need to be more upfront.  Especially since some readers do not read the conclusion, I suggest the authors make a note of the lack of sampling in the results section before any microbiota data is presented. Also all plots showing taxa should be moved to the supplement. This includes Figures 1D-J and Figures 4D-J.

We understand the criticism of reviewer 1. Thus, following both reviewers 1 and 2 comments, we re-organized the result and discussion sections to dampen the importance of the microbiota data and to warn the reader that the microbiota data need to be taken with caution due to the low n value. We removed the taxa plots from microbiota figures (now Figures 5 & 6) and referred to Tables S4 and S5 for family abundances data. We also modified the title (suggested by reviewer 2) and the limitation section.

Reviewer 2 Report

I appreciate the authors' efforts in reviewing the manuscript and for corrections.

After reviewing the manuscript closely and considering the the authors reply, my major concern abour the power and the lack of correction of cage effect has been even more substantiated. Because I believe this work is still important and worth publishing, I would suggest the authors shift the attention of the manuscript from the microbiome mainly to their other main readouts i.e. in the title already and also in the text. I would suggest adding a brief section with your power analyses to explain why you took the amount of animals you did. This will make it easier for the readers to be ok with the small n and as long as you state the lacking power as a limtation for microbiome analyses, the article would be much more sound. So concretely:

  • adapt title to shift attention on adiposity and adipose tissue phenotype/inflammation/gut barrier 
  • compile the microbiome effects and correlation into one smaller section at the end with subtitles, reducing the overall weight of the microbiome analysis, which I believe is not the strong suite of the paper. 

some minors still to correct: 

- cutadapt is not used for quality control. It can trimm according to certain quality criteria. why were sequences deleted which were between 380 and 500 bp?

- did you use 999 (typically used) or 9999 permutations for adonis? 

- rat housing: there was cohousing with 2 rats per cage: was this consistent throughout the experiment study design ( I would assume not, considering the sacrificing of animals in between?). did you see a cage effect? this should at least be controlled for in your betadiversity and while doing adonis. This could indeed be very impactful for the results you see considering the very low number of animals you have. if you reduce the microbiome part considerably as suggested above and clearly state the lack of correction of cage effect, that would be fine also. 

- you report results on family level. you state that Deseq2 analysis can only be performed to compare 2 groups, which was not relevant because your experimental design has 2 factors. This is news to me. Here is a direct excerpt from the DESEQ bioconductor page: 'In fact, DESeq2 can analyze any possible experimental design that can be expressed with fixed effects terms (multiple factors, designs with interactions, designs with continuous variables, splines, and so on are all possible). By adding variables to the design, one can control for additional variation in the counts'. So you might actually be able to do the tests in DESEQ, my worries at this point are the very small n in the groups, that and cohousing makes it hard to know if the results you show are robust. my suggestion: keep at family level and say you do this because of low power and confidence in higher ranks when merged together. 

- thank you for adding the methods used. Please do reference R ! it is still a software and it needs a reference next to it (it isn't referenced in statistical analyses).

- similarly Adonis function is from the vegan package. Please reference that in there .

Figure 1 and 4 D: is that really mean and SEM or Mean and SD? 

  • Limitations: please adress the issue of cage as described above, family level analyses. limitations pertaining todiscussion points I raise and you debated quite nicely should also be added in brief to the limitation section. 

Author Response

I appreciate the authors' efforts in reviewing the manuscript and for corrections.

After reviewing the manuscript closely and considering the the authors reply, my major concern abour the power and the lack of correction of cage effect has been even more substantiated. Because I believe this work is still important and worth publishing, I would suggest the authors shift the attention of the manuscript from the microbiome mainly to their other main readouts i.e. in the title already and also in the text. I would suggest adding a brief section with your power analyses to explain why you took the amount of animals you did. This will make it easier for the readers to be ok with the small n and as long as you state the lacking power as a limtation for microbiome analyses, the article would be much more sound. So concretely:

    adapt title to shift attention on adiposity and adipose tissue phenotype/inflammation/gut barrier
    compile the microbiome effects and correlation into one smaller section at the end with subtitles, reducing the overall weight of the microbiome analysis, which I believe is not the strong suite of the paper.

Following your comments, we re-organized the result and discussion sections to dampen the importance of the microbiota data and to warn the reader that the microbiota data need to be taken with caution due to the low n value. We also removed the taxa plots from microbiota figures (suggested by Reviewer 1) and deleted the OTU level analysis. We changed the title accordingly.

some minors still to correct:

- cutadapt is not used for quality control. It can trimm according to certain quality criteria. why were sequences deleted which were between 380 and 500 bp?

We changed the M&M accordingly to add trimming procedure. The sequences between 380 and 500 pb were kept and not deleted. This as a mistake in the previous version that is now fixed (Lines 160-161).

- did you use 999 (typically used) or 9999 permutations for adonis?

We used 9999 as stated in the method section.

- rat housing: there was cohousing with 2 rats per cage: was this consistent throughout the experiment study design ( I would assume not, considering the sacrificing of animals in between?). did you see a cage effect? this should at least be controlled for in your betadiversity and while doing adonis. This could indeed be very impactful for the results you see considering the very low number of animals you have. if you reduce the microbiome part considerably as suggested above and clearly state the lack of correction of cage effect, that would be fine also.

As already mentioned, we re-organized the results section and warned the reader on the low n value. The cohousing was consistent throughout the experiment. At 3 months, 2 cages per group were removed randomly so the rats will not stay alone in the cage, which could impact the animal stress and physiology. We also indicated that the cage effect was not taken into account in the statistical analysis (Line 171).

- you report results on family level. you state that Deseq2 analysis can only be performed to compare 2 groups, which was not relevant because your experimental design has 2 factors. This is news to me. Here is a direct excerpt from the DESEQ bioconductor page: 'In fact, DESeq2 can analyze any possible experimental design that can be expressed with fixed effects terms (multiple factors, designs with interactions, designs with continuous variables, splines, and so on are all possible). By adding variables to the design, one can control for additional variation in the counts'. So you might actually be able to do the tests in DESEQ, my worries at this point are the very small n in the groups, that and cohousing makes it hard to know if the results you show are robust. my suggestion: keep at family level and say you do this because of low power and confidence in higher ranks when merged together.

We followed the reviewer advice and removed the OTU level and Deseq2 analysis.

- thank you for adding the methods used. Please do reference R ! it is still a software and it needs a reference next to it (it isn't referenced in statistical analyses).

R is now cited (Line 159).

- similarly Adonis function is from the vegan package. Please reference that in there .

Added.

Figure 1 and 4 D: is that really mean and SEM or Mean and SD?

Yes, this is SEM as stated.

    Limitations: please adress the issue of cage as described above, family level analyses. limitations pertaining todiscussion points I raise and you debated quite nicely should also be added in brief to the limitation section

We addressed the various issues accordingly (lines 267-81).